# One-pot H/D exchange and low-coordinated iron electrocatalyzed deuteration of nitriles in D₂O to *α,β*-deuterio aryl ethylamines

Rui Li [1,4], Yongmeng Wu [1,4], Changhong Wang [1], Meng He[1], Cuibo Liu [1,2] ✉ & Bin Zhang [1,2,3] ✉

Developing a step-economical approach for efficient synthesis of *α,β*-deuterio aryl ethylamines (*α,β*-DAEAs) with high deuterium ratios using an easy-to-handle deuterated source under ambient conditions is highly desirable. Here we report a room-temperature one-pot two-step transformation of aryl acetonitriles to *α,β*-DAEAs with up to 92% isolated yield and 99% *α,β*-deuterium ratios using D₂O as a deuterium source. The process involves a fast *α*-C–H/C–D exchange and tandem electroreductive deuteration of C≡N over an in situ formed low-coordinated Fe nanoparticle cathode. The moderate adsorptions of nitriles/imine intermediates and the promoted formation of active hydrogen (H*) on unsaturated Fe sites facilitate the electroreduction process. In situ Raman confirms co-adsorption of aryl rings and the C≡N group on the Fe surface. A proposed H*-addition pathway is confirmed by the detected hydrogen and carbon radicals. Wide substrate scope, parallel synthesis of multiple *α,β*-DAEAs, and successful preparation of *α,β*-deuterated *Melatonin* and *Komavine* highlight the potential.

The drugs' metabolism and pharmacokinetic properties will be profoundly modified by the presence of the deuterium (D) atom due to its kinetic isotope effect[1–6]. Since the first deuterated drug, deutrabenazine (Austedo), has been approved by the US Food and Drug Administration (FDA)[7], tremendous efforts have been devoted to synthesizing and patenting D-labeled pharmaceuticals[8–16]. Aryl ethylamines can serve as both important drug molecules and ubiquitous scaffolds in biologically active compounds (Fig. 1a and Supplementary Fig. 1)[17–21]. Introducing D at the *α*- or *β*-position of aryl ethylamines has proven to produce a remarkable intensification of the blood pressure effect by *α,α*-bisdeuteriotryptamine or to significantly attenuate the metabolism of *α,β,β*-trideuteriodopamine via *β*-C–D bond cleavage compared with their unlabeled counterparts[22,23]. Furthermore, the inherently larger amount of D atoms is often necessary for the labeled compounds to be of practical use in pharmaceutical studies. In these regards, the incorporation of D at both *α*- and *β*-positions of

arylethylamines has a very great and urgent significance for further enhancing the drug's metabolic stability and bioactivity.

Transition metal-catalyzed C–H/C–D exchange is a prevailing strategy to install D atoms at the *α*- and *β*-positions adjacent to N-based moieties due to unrequired prefunctionalization (Fig. 1b)[24,25]. However, noble metals and harsh reaction conditions are often required to activate the inert C–H bond[24–27], leading to unsatisfactory regioselectivity, insufficient deuterium incorporation, and poor functional group tolerances. Meanwhile, the high cost of D₂ storage and transportation and the safety risk of D₂ usage are still two main concerns. Alternatively, reductive deuteration of *α*-deuterated aryl acetonitriles provides an attractive route to access *α,β*-DAEAs with high deuterated ratios[28–31]. However, the preparation and isolation of *α*-deuterated aryl acetonitriles are tedious and time-consuming. In addition, pyrophoric metal deuterides (e.g., LiAlD₄, NaBD₄) are always required with disposing of hazardous chemical wastes for the deuteration of C≡N.

[1]Department of Chemistry, Institute of Molecular Plus, School of Science, Tianjin University, Tianjin 300072, China. [2]Haihe Laboratory of Sustainable Chemical Transformations, Tianjin 300192, China. [3]Tianjin Key Laboratory of Molecular Optoelectronic Science, Collaborative Innovation Center of Chemical Science and Engineering, Tianjin 300072, China. [4]These authors contributed equally: Rui Li, Yongmeng Wu. ✉e-mail: cbliu@tju.edu.cn; bzhang@tju.edu.cn

**(a) Representative drugs and bio-active compounds based on aryl ethylamines**

| Tyramine | Tryptamine | $d_9$-Venlafaxine (SD-254) | SCH 12679 |
|---|---|---|---|
| [neuropromoter] | [neurotransmitter] | [anti-depressant] | [dopamine D2 receptors] |

**(b) Prevailing H/D-exchange methods**

○ noble metals and expensive D sources      ○ insufficient D incorporation

**(c) Reductive deuteration of $\alpha$-deuterated aryl acetonitriles**

○ harsh conditions ○ tedious operations ○ expensive D sources ○ hazardous chemical wastes

**(d) One-pot H/D exchange with electroreductive deuteration strategy (this work)**

● high D ratio      ● mild conditions      ● one pot synthesis      ● cheap D source and catalyst

**Fig. 1 | Schematic comparison of synthesis of deuterated aryl ethylamines.** **a** Representative examples of aryl ethylamine related drugs. Typical methods **b** and **c** and our proposed one-pot two-step deuteration strategy **d** for the synthesis of $\alpha,\beta$-deuterated aryl ethylamines.

These inherent factors bring about handling complexities, safety risks, and environmental issues. Given the ready availability of aryl acetonitrile substrates and easy H/D exchange of $\alpha$-C − H with D$_2$O under basic conditions, developing a one-pot transformation of aryl acetonitriles to $\alpha,\beta$-DAEAs will be more appealing from the step- and atom-economic point of view. Although reductive deuteration of aryl acetonitriles has been reported[32,33], current works are usually suffering from much lower D contents at $\beta$-carbon atoms using expensive reagents with strictly controlled anhydrous and inert conditions, and lacking mechanistic studies, restricting their practical applications. Therefore, it is highly desirable to search for a convenient and sustainable approach for a one-pot efficient conversion of aryl acetonitriles to highly deuterated $\alpha,\beta$-DAEAs using inexpensive and safe D$_2$O as a deuterated source under ambient conditions, and to unveil the underlying reaction mechanism.

Recently, electrocatalytic transformations have aroused increasing interest in the synthetic and catalytic fields[34–38]. Electrocatalytic hydrogenation by using H$_2$O as a hydrogen source is markedly appealing due to avoiding the need for handling high-pressure H$_2$ and other environmentally unfriendly hydrogen reagents. Thus, electroreduction of nitrate[39,40] and easily reducible organic groups, such as −NO$_2$, C − I, C = O, C = C, and C ≡ C has been extensively studied[34–38,41,42]. However, electroreduction of the C ≡ N skeleton has been rarely touched[43–46], which may be ascribed to its high bonding energy and strong coordination property leading to the deactivation of metal catalytic centers. Jiao and co-workers have recently made an advance in electrocatalytic hydrogenation of aliphatic nitriles to primary amines[47]. But, electrocatalytic hydrogenation and deuteration of aryl

acetonitriles have not been touched because the presence of aryl rings may cause different adsorption modes with altered reaction outcomes. In addition, D$_2$O is often adopted to confirm the hydrogenation mechanism in electrocatalytic hydrogenation reactions. We thus propose that the combination of $\alpha$-H/D exchange and electrocatalytic C ≡ N deuteration of aryl acetonitriles will be highly promising for achieving a one-pot synthesis of $\alpha,\beta$-DAEAs using D$_2$O. One of the main challenges is to select a suitable cathode to activate the aryl acetonitrile substrates. Very recently, engineering low-coordinate sites and vacancies into electrode materials can promote water splitting and govern the intrinsic activity and selectivity of hydrogenation reactions via optimizing adsorption[48–50]. Thus, designing an advanced material can be feasible to achieve a step-economical electrochemical conversion of aryl acetonitriles to $\alpha,\beta$-DAEAs using D$_2$O as the deuterium source.

Herein, we initially screened some typical cathode materials for the hydrogenation of aryl acetonitriles with H$_2$O as the hydrogen source, and Fe foil is the optimal one. Then, a carbon paper (CP)-supported low-coordinated Fe nanoparticles (LC-Fe NPs) cathode is designedly synthesized via in situ electroreduction of $\alpha$-Fe$_2$O$_3$/CP, which enables the one-pot deuteration of aryl acetonitriles to $\alpha,\beta$-DAEAs with high yields and good to excellent deuterated ratios using D$_2$O to replace H$_2$O (Fig. 1d). A series of in situ and ex situ characterizations confirm the conversion from $\alpha$-Fe$_2$O$_3$ to LC-Fe NPs. Theoretical results demonstrate that moderate adsorption of nitrile and imine intermediates and enhanced formation of active hydrogen are key to achieving high reaction efficiencies. This facile strategy can not only lead to the highly selective synthesis of diverse $\alpha,\beta$-DAEAs, but also be

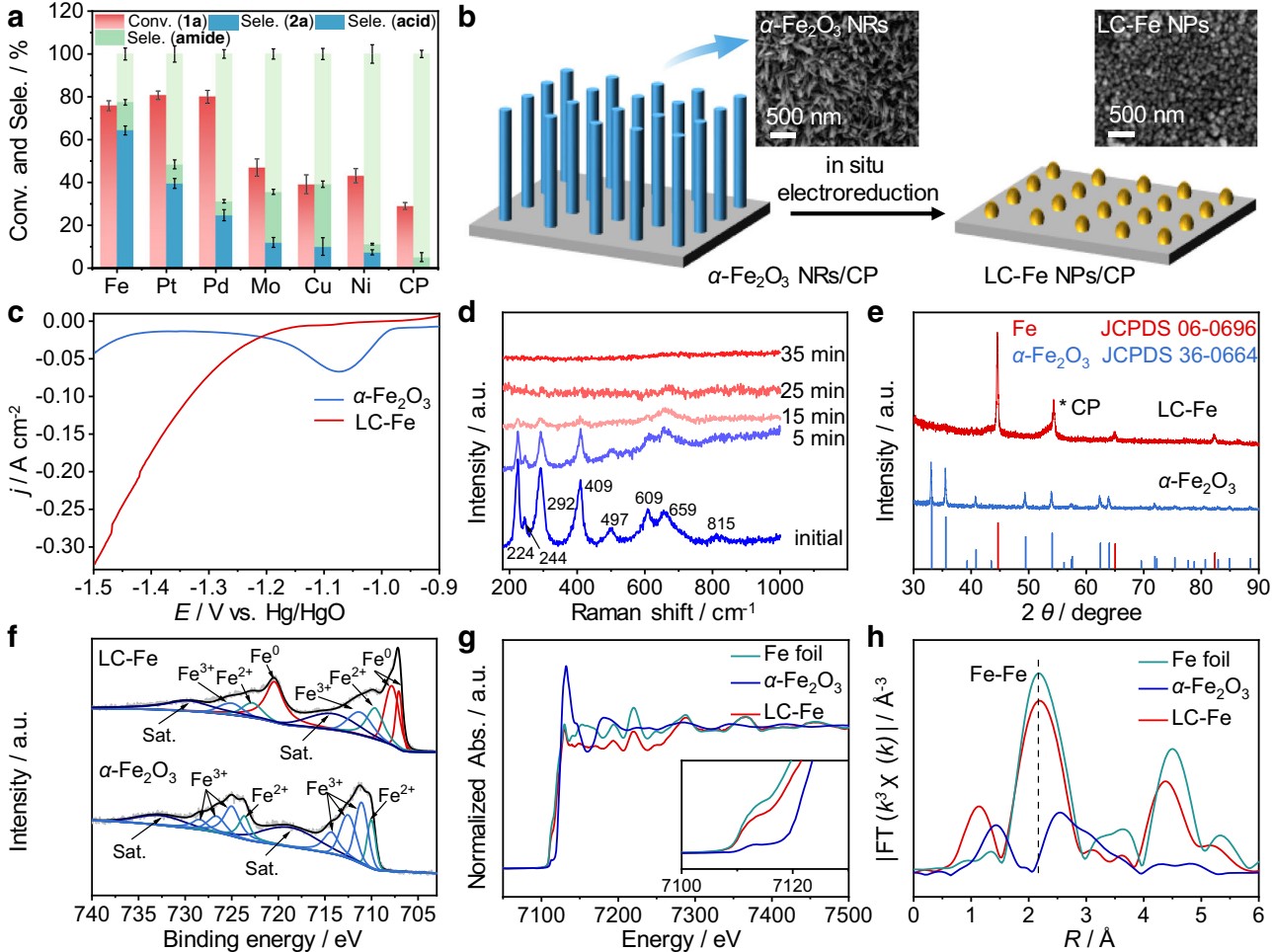

**Fig. 2 | Synthesis of CP-supported LC-Fe NPs via electroreduction of α-Fe₂O₃ NRs. a** Comparison results of converting **1a** to **2a** over different cathode materials. **b** A schematic illustration of the electroreduction conversion from α-Fe₂O₃ NRs to LC-Fe NPs and their corresponding SEM images. **c** LSV curves of α-Fe₂O₃ NRs and as-prepared LC-Fe NPs at a scan rate of 10 mV s⁻¹ in 1.0 M KOH solution. **d** Time-dependent in situ Raman spectra of α-Fe₂O₃ NRs collected at −1.9 V in 1.0 M KOH; **e** XRD pattern of α-Fe₂O₃ NRs and LC-Fe NPs. **f** XPS spectra of α-Fe₂O₃ NRs and LC-Fe NPs (the sputtering time of LC-Fe NPs is 120 s). **g** In situ Fe K-edge XANES spectra and **h** EXAFS spectra of LC-Fe NPs, α-Fe₂O₃ NRs, and Fe foil. Error bars correspond to the standard deviation of three independent measurements. Reaction conditions for **a**: **1a** (0.1 mmol), the mixed 1.0 M KOH/dioxane (3:1 v/v, 8 mL), −1.4 V vs. Hg/HgO, RT, 8 h.

further developed to synthesize *d₄-Melatonin* with hormone modulation activity and to label the natural product *d₄-Komavine* with deuterium, showing the practical utilities.

## Results

### An alkaline electrolyte and a low-coordinated Fe electrocatalyst

First, the co-existence of the aryl ring and strong electron-withdrawing cyano (CN) group makes α-C − H bond of aryl acetonitriles highly active, leading to a facile H/D exchange with D₂O in the presence of a base. We observe a quick and complete conversion of 0.1 mmol of *p*-methoxyphenylacetonitrile (**1a**, a model substrate) to its α-deuterio analog in a mixed D₂O/dioxane solution with the assistance of KOH or K₂CO₃ (Supplementary Fig. 2), ensuring a high α-deuterated content. In addition, electroreduction in an alkaline medium is commonly conducive to maintaining the high stability of electrodes and to achieving a satisfying Faradic efficiency (FE) of the targeted product via inhibiting competitive hydrogen evolution reaction (HER)[51]. On the basis of these considerations, we use KOH or K₂CO₃ as a base for implementing the one-pot transformation of aryl acetonitriles to α,β-DAEAs using D₂O as the deuterated source.

Second, the α-C − H/D exchange is a fast but not a catalytic process, and thus efficient reductive deuteration of the CN group

becomes crucially significant to expediently synthesize α,β-DAEAs from aryl acetonitriles. For saving the use of D₂O, we select electrocatalytic hydrogenation of **1a** with H₂O to determine the optimal catalyst and to study the related performances and reaction mechanism thereinafter. Electrochemical experiments are carried out in a divided H-type cell (Supplementary Fig. 3). Commonly, a mixed solution of 1.0 M KOH/dioxane (3:1 v/v, 8 mL) is used to better dissolve **1a**. We screen a series of cathode materials adopting the same applied potential of −1.4 V vs. Hg/HgO (potentials in this work are all referred to the Hg/HgO unless otherwise stated) to provide sufficient active hydrogen (H*) via H₂O electrolysis for **1a** electroreduction. Among all the tested materials (Supplementary Note 1), Fe foil shows the highest selectivity of 4-methoxyphenylethylamine (**2a**) with near 76% conversion of **1a** during an 8 h potentiostatic electrolysis (Fig. 2a, Supplementary Table 1, Supplementary Fig. 4, and Supplementary Notes 2, 3). Although similar conversions of **1a** are observed over Pt and Pd compared with the Fe foil, **2a** selectivities are much lower, mainly due to the formation of amide and carboxylic acid byproducts (60 and 75% selectivities of byproducts over Pt and Pd, respectively). This may be attributed to their excellent activities for the hydrogen evolution reaction (HER)[52], thus inhibiting the hydrogenation of **1a**. In addition, the strong coordination between organic nitrogen species (e.g., nitrile

substrate, imine intermediate, amine product) and Pt or Pd may be another reason for the low selectivity of **2a**. Mo, Ni, and Cu cathodes demonstrate very weak activity and low selectivity of **2a** toward the **1a** electroreduction, while CP is almost inert. We speculate that the poor performance of Mo, Cu, and CP is ascribed to their much stronger or weaker hydrogen binding energies[52]. And the higher electronegativity of Ni causes the harder release of **2a** from Ni, accounting for its worse hydrogenation performance than Fe. Therefore, Fe will be a good candidate for the electrocatalytic hydrogenation of **1a** with $H_2O$. In addition, the hydrolysis of nitriles can easily proceed under either acidic or alkaline conditions to produce byproducts, including amides, carboxylic acids, and their salts[53]. In this regard, designing more efficient materials for accelerating the reductive hydrogenation of the CN group is urgently needed. Meanwhile, tuning the electron structure of an electrocatalyst is often reported to improve its electrochemical performance[54]. Thus, modifications of Fe-based materials are extremely required to further enhance the reaction activity and selectivity toward the **1a** electroreduction. Furthermore, the low-coordinated metal sites with tuned electron structure or coordination environment will help to modulate the adsorption of reactants or intermediates and the affinity of H* from $H_2O$ electrolysis, thus influencing reaction efficiency and product distributions[48,49,55,56]. Therefore, we aim at synthesizing a low-coordinated Fe electrocatalyst to more effectively activate the CN and related intermediate and promote the formation of H*, hence facilitating electrocatalytic hydrogenation of aryl acetonitriles for the efficient synthesis of α,β-DAEAs with inhibiting the possible amide and carboxylic acid hydrolysates.

## Synthesis and characterizations of a low-coordinated Fe nanoparticle/CP electrocatalyst

Electroreduction offers a promising technique to convert material precursors into highly active electrocatalysts. Conductive substrate-supported electrodes can prevent the usage of insulating binders, thus exposing more active sites and improving electrical conductivity. Generally, oxides-derived metals own low-coordinated surfaces[55,56]. The Pourbaix diagram of Fe (Supplementary Fig. 5) can guide us in the synthesis of LC-Fe. So, CP-supported low-coordinated LC-Fe is synthesized through a facile electroreduction treatment of α-$Fe_2O_3$ nanorod precursors[57] in a 1.0 M KOH electrolyte at −1.9 V (Fig. 2b, CP support of LC-Fe/CP and α-$Fe_2O_3$/CP is omitted for a brief description hereinafter). The scanning electron microscopy (SEM) images, X-ray diffraction (XRD) pattern, and X-ray photoelectron spectroscopy (XPS) spectra suggest the successful preparation of α-$Fe_2O_3$ nanorods (NRs) (Fig. 2b, e–f, Supplementary Fig. 6, and Supplementary Note 4). After a period of electrolysis (Fig. 2b), the reduction-induced oxygen stripping can decrease the size of the NRs and cause the transformation of NRs to small nanoparticles (NPs). In the linear sweep voltammetry (LSV) curves (Fig. 2c), an obvious reduction peak of α-$Fe_2O_3$ disappears after the electroreduction treatment, which demonstrates the full conversion of the surface layer in α-$Fe_2O_3$ to Fe(0). Electrochemical in situ Raman spectra also record the transformation process (Fig. 2d). The spectrum of the initial sample shows that the peaks located at around 224 and 497 $cm^{-1}$ correspond to the $A_{1g}$ of α-$Fe_2O_3$, and the peaks of 244, 292, 409, 609, and 815 $cm^{-1}$ are assigned to the $E_g$ modes. In addition, the mode $E_u$ of 659 $cm^{-1}$ is the result of the disorder and grain size of α-$Fe_2O_3$[58]. When the electrolysis begins, the intensity of these characteristic peaks gradually weakens and finally vanishes after 35 mins, further confirming the complete transformation from α-$Fe_2O_3$ surface to Fe(0). All the diffraction peaks in the XRD pattern are indexed to the (110), (200), and (211) planes of cubic Fe (JCPDS NO. 06-0696, Fig. 2e). Electrochemical in situ XRD is also carried out to monitor the phase transformation of α-$Fe_2O_3$ (Supplementary Fig. 7). In the XRD patterns, the characteristic peaks of α-$Fe_2O_3$ disappear and the peaks belonging to the Fe(OH)$_2$ become prominent after a period of electrolysis. With the electrolysis going on, the peaks of Fe(OH)$_2$

vanish, and the characteristic peaks corresponding to cubic Fe raise, which is the final and stable phase of the reduced sample. Thus, these XRD results suggest the reduced conversion process of α-$Fe_2O_3$ → Fe(OH)$_2$ → Fe. Furthermore, in the Fe 2p XPS spectra (Fig. 2f), two prominent peaks located at 707.1, 707.8, and 720.4 eV are belonged to the Fe $2p_{3/2}$ and Fe $2p_{1/2}$, respectively. These peaks demonstrate negative shifts compared with those of α-$Fe_2O_3$, suggesting the formation of metallic Fe(0)[59]. And, the small deconvoluted peaks at approximately 709.7, 711.4, 722.8, and 725.2 eV are assigned to the oxidized Fe owing to the inevitable oxidation of the Fe surface during the test[60].

In situ X-ray absorption spectroscopy (XAS) is performed to understand the accurate electronic configuration and local coordination environments. The Fe K-edge X-ray absorption near-edge structure (XANES) of LC-Fe exhibits similar features to that of the Fe foil (Fig. 2g), suggesting that the reduced sample mainly consists of metallic Fe. However, the absorption edge position of LC-Fe is located between those of α-$Fe_2O_3$ and Fe foil, demonstrating a higher valence state of LC-Fe. This may be ascribed to the existence of low-coordinated unsaturated Fe sites after electroreduction. The Fourier-transformed $k^3$-weighted extended X-ray absorption fine structure (EXAFS) (Fig. 2h) shows the new arise of the Fe−Fe path in the reduced sample (around 2.3 Å), while the average Fe−Fe coordination number is lower than that of Fe foil (Supplementary Fig. 8 and Supplementary Table 2). Whereas, the peak at around 1 Å may be ascribed to the Fe−C path caused by the interaction between Fe and the CP support[61]. These results indicate that α-$Fe_2O_3$ nanorods experience morphological and structural evolution during the electrochemical reduction process, and the in situ formed LC-Fe possessed low-coordinated sites, benefiting the electroreductive hydrogenation/deuteration of aryl acetonitriles with $H_2O$/$D_2O$.

## Electrocatalytic hydrogenation of 1a with $H_2O$ over the LC-Fe cathode

After obtaining the low-coordinated Fe cathode, we examine its performance on the electrocatalytic hydrogenation of **1a** with $H_2O$ under ambient reaction conditions. The LSV curve displays an apparent increase of the current density between −1.1 and −1.3 V after adding 0.1 mmol of **1a** into the cathodic cell, implying an easier reduction of **1a** than the HER in this potential range (Fig. 3a). At potentials more negative than −1.3 V, the HER becomes the dominant reaction, which may be caused by the mass transport limitation of **1a** to the electrode interface[43,44,47]. Potential screening results reveal that **1a** can be electroreduced from −1.1 V, corresponding to the LSV result. Although there is about 40% conversion of **1a** at −1.0 V, the products are a mixture of amide and carboxylic acid hydrolysates rather than the desirable primary amine **2a**. The conversion of **1a** can reach 97% with up to 97% selectivity of **2a** in the potential range from −1.15 to −1.3 V (Fig. 3b). In sharp contrast, Fe foil needs more negative potentials and exhibits much worse activity toward **1a** electroreduction. For example, the **2a** selectivity over as-prepared LC-Fe is approximately 21.6 times at −1.25 V and 4.06 times at −1.3 V than those over corresponding Fe foil, respectively. These comparison results reflect the intrinsic high activity of LC-Fe, thus rationalizing our speculations. Time-dependent experiments in Fig. 3d demonstrate that **1a** can be nearly hydrogenated to **2a** with 97% selectivity in about 6 h at −1.2 V. Note that, the yield of **2a** has already reached 61% within the first hour, and the calculated production rate of **2a** is 0.061 mmol $cm^{-2}$ $h^{-1}$. In addition, electrocatalytic hydrogenation of **1a** can also be carried out under galvanostatic conditions (Fig. 3e), and only 3.5 h is required when applying a 40 mA $cm^{-2}$ of current density to achieve a comparable result with that at −1.2 V, demonstrating good operability of our method. Furthermore, the LC-Fe electrocatalysts can be repeatedly used for five runs with maintaining excellent conversion of **1a** and high selectivity of **2a** (Fig. 3f). A slight decrease of **2a** selectivity for the sixth

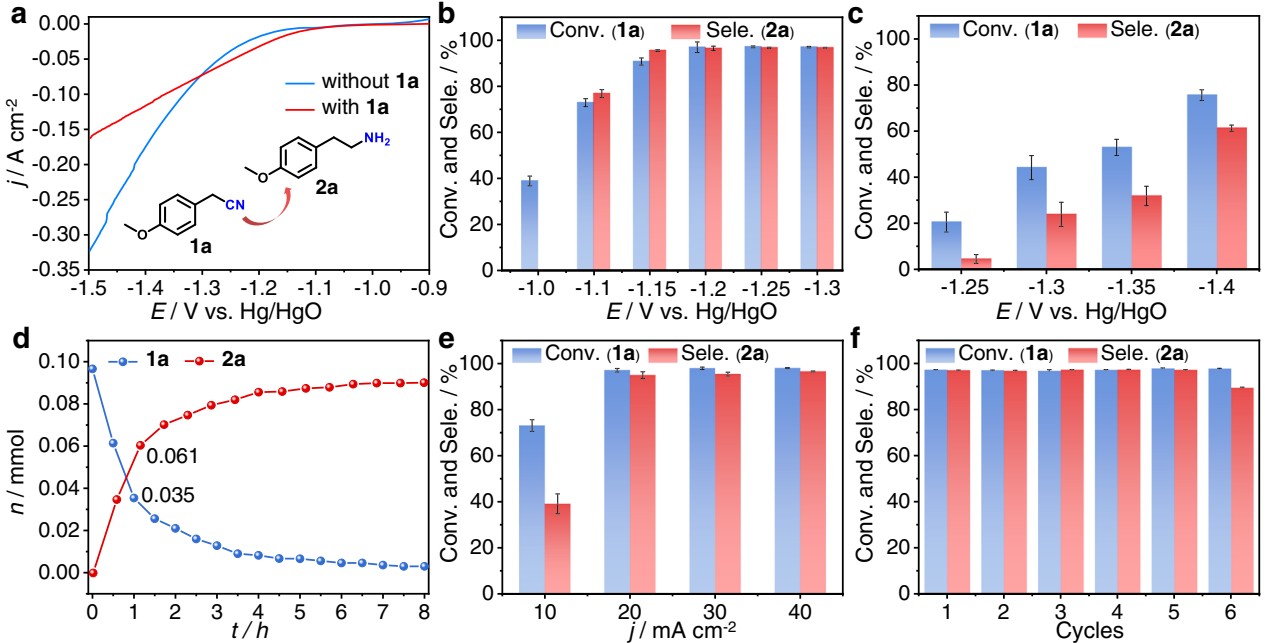

**Fig. 3 | Performances of the electrocatalytic hydrogenation of 1a over the LC-Fe cathode. a** LSV curves of LC-Fe at a scan rate of 10 mV s⁻¹ in 1.0 M KOH/dioxane (3:1 v/v, 8 mL) mixed solution with and without **1a**. Potential-dependent conversion (Conv.) of **1a** and selectivity (Sele.) of **2a** over **b** LC-Fe and **c** Fe Foil, respectively. **d** Time-dependent **1a** Conv. and **2a** Sele. over LC-Fe. **e** Galvanostatic electrolysis of **1a** over LC-Fe. **f** Cycle-dependent **1a** Conv. and **2a** Sele. over LC-Fe. Error bars correspond to the standard deviation of three independent measurements. Reaction conditions: **1a** (0.1 mmol), a mixed solvent of 1.0 M KOH/dioxane (3:1 v/v, 8 mL), RT, **b** and **c** at different potentials, 8 h; **d** at −1.2 V, 8 h; **e** at different current densities, 8 h; **f** at −1.2 V, 8 h.

time may be mainly ascribed to the stripping of some catalysts from the CP support that slows down the **1a** hydrogenation process and enhance the hydrolysis of **1a**.

## Mechanistic studies

Generally, hydrogenation of nitriles to primary amines is through a two-step process involving an imine (C = N) intermediate[46,47,62,63], and we investigate the reaction mechanism by combining control experiments with theoretical calculations. First, to determine whether hydrogenation of **1a** occurs in the bulk solution or on the surface of the electrode, we employ a long-chain thiol as the capping reagent to modify the LC-Fe cathode before the electrolysis begins. Nearly no **2a** is detected after the modification of LC-Fe, and we observe a small amount of **1a** converting to the carboxylic acid hydrolysate instead (Fig. 4a). This may demonstrate that the electrocatalytic hydrogenation of **1a** mainly proceeds on the LC-Fe surface. Second, in situ Raman spectroscopy is employed to provide crucial information on the adsorption modes of **1a** on the LC-Fe surface and to unveil the details of the hydrogenation process. The characteristic Raman band at 2252 cm⁻¹ belongs to the C ≡ N vibrational ($\nu_{C\equiv N}$) mode of pure **1a**, and two peaks located at 1589 and 1615 cm⁻¹ are ascribed to the vibration of C = C ($\nu_{C=C}$) bonds of the benzene ring (Fig. 4b, middle)[46,64,65]. When the Raman tests are conducted in the presence of the LC-Fe cathode at −1.2 V (Fig. 4b, up), the main $\nu_{C\equiv N}$ mode remains almost unchanged but a small peak at around 2218 cm⁻¹ appears. This may be due to the adsorption of the CN group on LC-Fe by the lone electron pair of N atoms at an uncertain angle rather than a vertical angle[46,64]. Meanwhile, for the $\nu_{C=C}$ modes, the peak intensity at 1589 cm⁻¹ increases while that of position 1615 cm⁻¹ decreases, indicating the adsorption of the benzene ring on the LC-Fe surface[64]. Additionally, a new peak at around 1614 cm⁻¹ corresponding to the vibration of the C = N ($\nu_{C=N}$) bond appears after 10 mins, and its intensity firstly increases and then decreases as the electrolysis goes on. This may indicate an imine-like intermediate involved in the hydrogenation process[19]. Surprisingly,

after adopting a more negative potential of −1.3 V (Fig. 4b, bottom), we observe a remarkable red-shift of the $\nu_{C\equiv N}$ mode (2071 vs. 2252 cm⁻¹) and a similar change of the $\nu_{C=C}$ modes compared with those at −1.2 V. The large $\nu_{C\equiv N}$ vibration differences may be ascribed to the oriented parallel adsorption of the CN group on the LC-Fe surface at more negative potentials. Therefore, rehybridization of the C ≡ N triple leads to both C and N atoms bonding to Fe sites, as reported by the Tian group[64]. These two distinct adsorption models of **1a** over LC-Fe at relatively positive and negative potentials enable the hydrogenation of aryl acetonitriles in a wide range of potentials with high yields and selectivity. Third, LSV studies of LC-Fe are performed to investigate the reduction behavior of **1a** under different conditions (Supplementary Fig. 9). We see a small reduction peak between −1.1 and −1.3 V in the LSV curve after adding 0.1 mmol of **1a** into anhydrous *N,N*-dimethyl-formamide (DMF) solvent by using tetrabutylammonium tetra-fluoroborate (TBAPF₄) as the electrolyte, revealing the electron transferring from the LC-Fe cathode to **1a**[46,47]. The current density increases obviously from −1.0 V after further adding 100 μL of H₂O to the reaction system, implying the promotional role of H₂O for this electrocatalytic hydrogenation of aryl acetonitriles. And, it was revealed that electroreduction of imine began with an electron transferring from the cathode to imine[62].

Density functional theory (DFT) calculations are performed to unveil the high-performance origin of the electrocatalytic hydrogenation of aryl acetonitriles over LC-Fe (phenylacetonitrile **1f** was selected to simplify the calculation, Supplementary Note 5). Because adsorption and desorption are vital for heterogeneous transformations, we primarily calculate the adsorption energies ($E_{ads}$) of substrate **1f**, imine intermediate **2f_Int**, and product **2f** on the catalysts' surfaces. After optimization of adsorption configuration (Supplementary Fig. 10), **1f** and **2f_Int** have prioritized adsorptions on LC-Fe via co-adsorption of aryl ring, C ≡ N and C = N groups (Fig. 4c), corresponding to the in situ Raman results. However, their $E_{ads}$ are much lower than those on Fe foil (−2.49 vs. −2.75 eV and −2.77 vs. −2.83 eV). It is reported

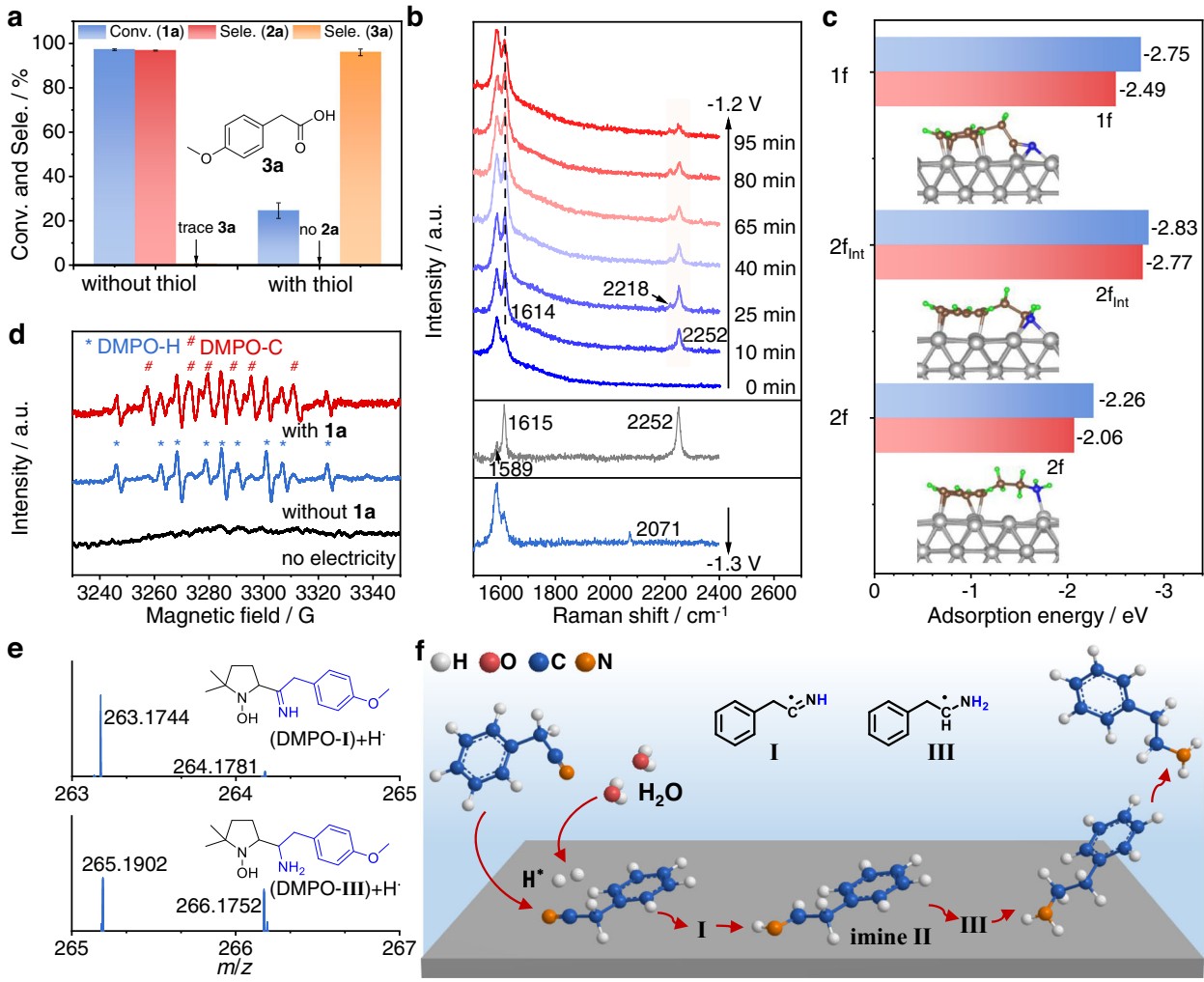

**Fig. 4 | Combined control experiments with theoretical calculations for mechanistic studies. a** Compared results of electrocatalytic hydrogenation of **1a** over LC-Fe and 1-dodecanethiol ($2 \times 10^{-3}$ mmol) modified LC-Fe, respectively, under standard conditions. **b** In situ Raman tests in a mixed 1.0 M KOH/dioxane (3:1 v/v, 8 mL) solution for electrocatalytic hydrogenation of **1a** (0.1 mmol) over LC-Fe at −1.2 and −1.3 V, respectively. **c** Comparisons of $E_{ads}$ of **1 f**, **2 f$_{Int}$**, and **2f** on Fe foil (blue) and LC-Fe (red), respectively (insert: the stable adsorption modes of **1 f**, **2 f$_{Int}$**, and **2f**). **d** Electron paramagnetic resonance trapping for hydrogen (*) and carbon (#) radicals over LC-Fe. **e** HR-MS analysis of the spin-trapping experiment of carbon radicals during **1a** electroreduction with $H_2O$ by using DMPO as a trapping agent. **f** A proposed reaction mechanism.

that moderate adsorption of nitriles and intermediates on the electrode surface is necessary for electrochemical hydrogenation of nitriles because the strong coordination ability of C≡N and C=N groups will slow down the hydrogenation process or deactivate the catalysts[46,47]. Thus, we speculate the weakened adsorptions of nitriles and imine intermediates on LC-Fe will certainly contribute to facilitating the hydrogenation process. The adsorption of **2 f** on LC-Fe is much weaker compared with that of **1 f** and **2f$_{Int}$**, conducive to regenerating the active sites and avoiding deactivation of LC-Fe. In addition, we observe a stronger N 1*s* signal in the N 1*s* XPS spectrum when treating LC-Fe with **1a** under our reaction conditions for 1 h (Supplementary Fig. 11). However, nearly no adsorption of **2a** on LC-Fe is observed. This may suggest the stronger adsorption of nitrile substrates on the LC-Fe surface than that of amine products, consistent with the DFT results. Furthermore, the Gibbs free energy ($\Delta G_{H^*}$) for H* formation on LC-Fe is more negative than that on Fe foil (−0.69 vs. −0.56 eV, Supplementary Fig. 12). This indicates a favorable production of H* via $H_2O$ activation on LC-Fe, which is also essential for speeding up the hydrogenation of nitriles and improving the amine selectivity. Overall, the DFT results suggest that LC-Fe leads to the moderate

binding affinity of nitriles/intermediate imines and promotes the formation of H*, thus helpful to the hydrogenation of aryl nitriles.

On the basis of the above experimental and theoretical results, a possible reaction mechanism is proposed (Fig. 4f and Supplementary Fig. 13). Aryl acetonitrile adsorbs on the LC-Fe surface via both the aryl ring and CN group. After the electrolysis starts, **1 f** accepts an electron to produce the arylacetonitrile radical anion, which subsequently abstracts a proton from $H_2O$, generating the carbon radical intermediate **I**. Then, the **I** couples with an adsorbed H* on the nearby Fe site that generates via $H_2O$ electrolysis delivering the half-hydrogenated imine **II** adsorbed on the LC-Fe surface. We can't detect imine **II** due to its highly active, but we trap the phenylmethanimine by benzylamine in electrocatalytic hydrogenation of benzonitrile with $H_2O$ (Supplementary Fig. 14), as in a similar work recently reported by Beller et al[66]. This result further proves the formation of imine intermediate during electrocatalytic hydrogenation of nitriles with $H_2O$. Further hydrogenation of imine to the full-hydrogenated amine product may experience similar processes to the formation of imine, and a carbon radical **III** may be also involved. The possible H* (also referred hydrogen radicals) and carbon radicals involved in this

reaction are detected by the electron paramagnetic resonance (EPR) measurements using 5,5-dimethyl-1-pyrroline-N-oxide (DMPO) as the trapping agent (Fig. 4d and Supplementary Note 6). The calculated g-value of 2.0043 with the hyperfine splitting coupling constants of $\alpha_N$ = 16.3 G and $\alpha_H$ = 22.2 G allows for the assignment of the signal to a spin adduct of DMPO-H (marked by *)[46,67,68]. Whereas, for the signals of DMPO-C (marked by #), the g-value, $\alpha_N$, and $\alpha_H$ are calculated as 2.0046, 15.7 G, and 22.3 G, respectively. Furthermore, the HR-MS data of DMPO-H and DMPO-C adducts further validate the involved hydrogen and carbon free radicals in the hydrogenation process of **1a** (Fig. 4e, Supplementary Fig. 15, and Supplementary Note 7). Due to the weak adsorption of **2a** on LC-Fe, it easily desorbs to leave the active sites regenerating for the next reaction cycle.

### Methodology universality and utility

Next, we first investigate the generality of our method for the synthesis of functionalized aryl ethylamines via electrocatalytic hydrogenation of aryl acetonitriles with $H_2O$ over the LC-Fe cathode (Supplementary Notes 3, 8, and 9). As seen in Supplementary Table 3, a wide range of aryl acetonitrile substrates are amenable to our strategy, giving rise to the corresponding aryl ethylamines in good to excellent yields with good functional group tolerance. Interestingly, chemoselective hydrogenation of the CN group can be achieved in the presence of reducible C − Cl and C − Br bonds (**2c** and **2d**), which may be due to the specific CN adsorption on the LC-Fe surface. However, the compatibility of more readily reduced C ≡ C and C = C bonds, which are usually difficult to survive in thermo-, photo-, and electrocatalytic hydrogenation of nitriles, remains a challenge (Supplementary Fig. 16). In addition, our strategy can also be expanded to the hydrogenation of aryl nitriles to produce benzylamines with high yields (**2i-l**), verifying the remarkable flexibility of our methodology. These results of electrocatalytic hydrogenation of nitriles greatly encourage us to validate the feasibility to synthesize $\alpha,\beta$-DAEAs with $D_2O$ because of the deuterium isotope effect causing changes in the drugs' metabolic properties.

The electroreductive deuteration of nitriles is then performed using $D_2O$ to replace $H_2O$ (Supplementary Notes 3, 8, and 9). As expected, the $\alpha,\beta$-DAEAs of drug-related building blocks (Supplementary Fig. 17) bearing electron-withdrawing and -donating groups at different positions of aryl rings can be obtained with 61-92% isolated yields (Fig. 5a, Supplementary Fig. 18). $\alpha$-C − H/D exchange has negligible influences on reductive deuteration of the CN group, and good to excellent D ratios are observed at both $\alpha$ and $\beta$ positions. The challenging C − Cl and C − Br bonds, which usually cleavage in thermocatalysis or via direct cathodic reduction over a CoP cathode[68,69], retain well and then provide good opportunities for fabricating complex deuterated molecules (**2s-t**, **2x**, **2 y**, and **2ae**). Especially, compounds **2p-q and 2aaa** acting as important drugs for treating neuropsychiatric disorders are synthesized with excellent D incorporations, which are often obtained through multiple procedures with expensive deuterated reagents in the reported methods[28,29]. $\alpha$-methyl-substituted aryl nitriles (**2ag**) and heterocycle containing (**2ah-ai**, and **2aaa-aab**) are also good candidates to deliver $\alpha,\beta$-DAEAs with good efficiencies. In addition, our strategy can be applied to alkyl nitriles with long-chains and various functionalized aryl nitriles, affording the corresponding $\alpha$-deuterated amines with high yields and good to high D ratios (**2aj-ak** and **2al-az**). Moreover, a distinct advantage of our method is exemplified by the parallel synthesis of multiple $\alpha,\beta$-DAEAs with comparable yields and deuterated ratios without altering the scale of the reaction setup (Fig. 5b).

Impressively, the as-obtained **2z** can be used to synthesize D-containing tetrabenazine with D atoms at the N-heterocycle (Fig. 6a), which may provide a useful complement to deutetrabenazine bearing −$OCD_3$ moieties for chorea associated with Huntington disease[10,20]. Furthermore, $d_4$-Melatonin with hormone modulation activity and the

deuterated natural product $d_4$-Komavine are finally accessed by further derivation of **2aab** and **2aaa** building blocks (Fig. 6b, c and Supplementary Notes 10, 11)[21], offering good opportunities to enhance their pharmacokinetic and pharmacodynamic properties. These satisfactory results reveal that our strategy can provide a powerful tool for deuterium labeling in organic synthesis and drug production.

## Discussion

Given the crucial importance and usefulness of $\alpha,\beta$-DAEAs for the development of deuterated drugs, searching for a mild, efficient, and step-economic synthetic strategy by using a cheap deuterated reagent is urgently needed. This work displays a one-pot two-step reaction procedure including $\alpha$-C − H to $\alpha$-C − D transformation and tandem electrocatalytic reductive deuteration of the C ≡ N bond over the designed LC-Fe cathode in an alkaline $D_2O$ solution. The one-pot consecutive protocol avoids the separation of $\alpha$-deuterated aryl acetonitriles in the traditional synthesis of $\alpha,\beta$-DAEAs. And, the reductive deuteration of CN ensures the synthesis of deuterated amines with high D content and selectivity compared with transition metal-catalyzed C − H/C − D exchange approaches. To achieve an effective synthesis of $\alpha,\beta$-ADPAs from aryl acetonitriles using $D_2O$ as the D source, a strong base (e.g., KOH, $K_2CO_3$) is extremely required to realize a fast H/D exchange with a high D introduction. The abundance in the earth's crust and low toxicity make Fe an ideal candidate for catalysis. The better performance of Fe on electrocatalytic hydrogenation of nitriles than that of other cathodes may be ascribed to its moderate binding energy of hydrogen and easy desorption of amines from the Fe surface due to the low electronegativity of Fe. In addition, moderate adsorptions of nitrile substrates and imine intermediates on the LC-Fe surface are significant to facilitate the reduction process, which follows a Sabatier rule.

Furthermore, the scale-up fabrication of $\alpha,\beta$-DAEAs is of paramount importance in the pharmaceutical industry. Flow chemistry holds great promise for the practical production of targeted products on a large scale. Thus, the design of a flow reactor is highly desirable to further improve the temporal and spatial availability of $\alpha,\beta$-DAEAs, which will be conducive to promoting drug discovery and development. Moreover, this methodology enables the one-pot synthesis of a series of functionalized $\alpha,\beta$-DAEAs from readily available aryl acetonitriles and low-cost $D_2O$ with high tolerance to the reducible C − Cl and C − Br bonds, but more fragile C ≡ C and C = C bonds are hard to survive. This may be fixed by further modifications of the catalyst to intrinsically improve the specific adsorption of CN while inhibiting the adsorptions of other reducible functional groups.

Moreover, the radical pathways are usually involved in aqueous electrocatalytic hydrogenation reactions (e.g., hydrogenation of nitrate, carbon dioxide, and organics)[42–50,55–57,67–69]. In our work, the key hydrogen, $sp^2$-C, and $sp^3$-C free radical intermediates related to the hydrogenation of nitriles with $H_2O$ are detected by combined EPR and HR-MS tests, thus a hydrogen radical addition or coupling pathway is proposed, which is distinct from the prevailing thermocatalytic hydrogenation of nitriles using $H_2$ as the hydrogen source, where the $H^+$ and $H^-$ are involved[62].

In summary, we demonstrate an expedient and economic one-pot two-step strategy for converting aryl acetonitriles to $\alpha,\beta$-DAEAs with high yields and outstanding deuterated efficiency. The whole process involves a fast $\alpha$-C − H/D exchange with a subsequent efficient electroredcutive deuteration of CN over a highly active low-coordinated Fe electrocatalyst by using $D_2O$ as a deuterated source. Mechanistic studies show that moderate adsorptions of nitrile substrates and imine intermediates and enhanced formation of active hydrogen are two crucial factors for the $\alpha,\beta$-DAEAs synthesis with high selectivity and efficiency. Apart from aryl acetonitriles, our electrocatalytic deuteration method is also well applied to aryl and alkyl nitriles and can be further developed to multiple syntheses of different $\alpha,\beta$-DAEAs in one

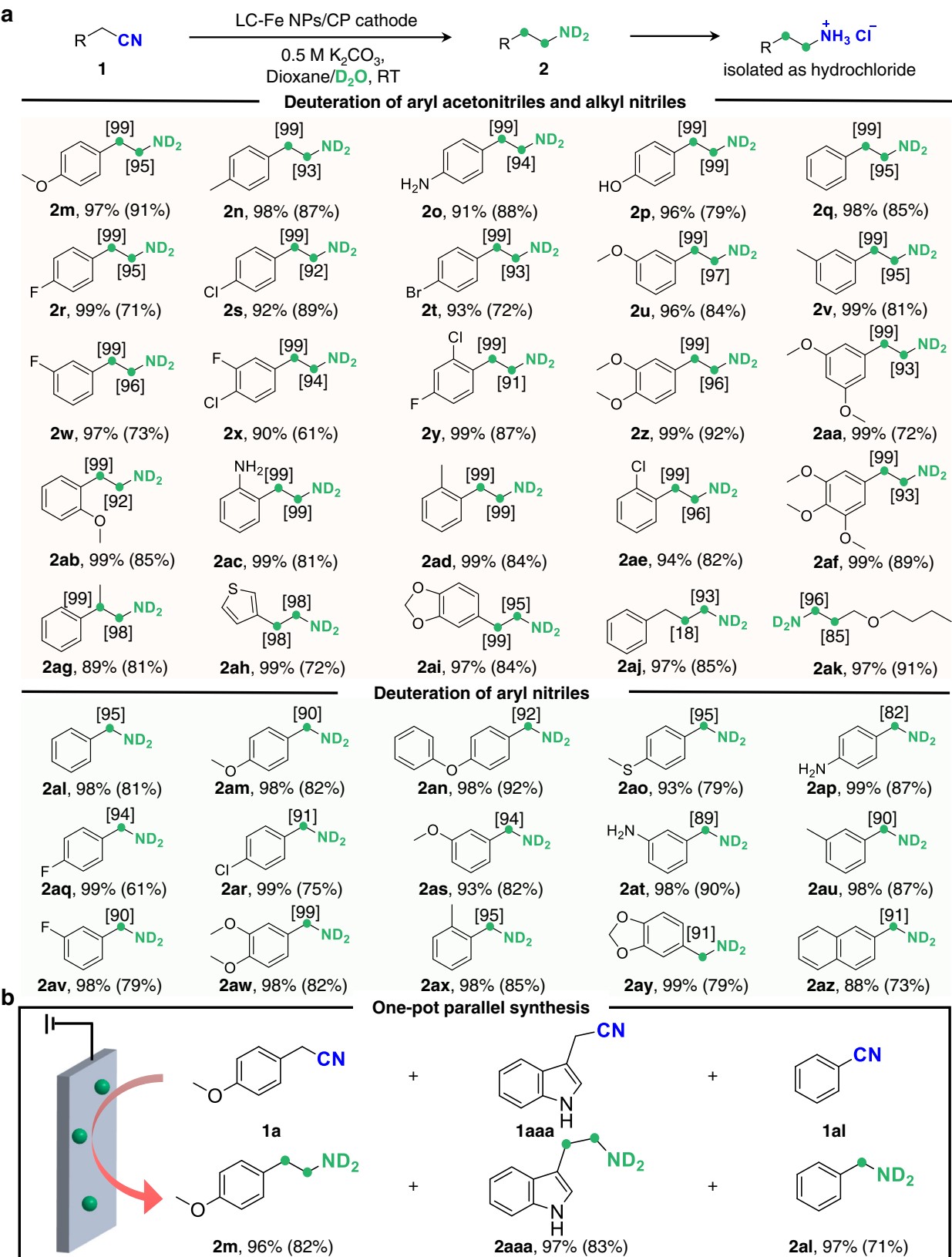

**Fig. 5 | The universality of our electroreductive deuteration method. a** α,β-DAEAs and α-deuterated amines containing useful functional units accessed by LC-Fe (isolated yields of the hydrochloride of primary amines are reported in the parentheses and deuterium ratios are presented in the brackets). Reaction conditions for the synthesis of α,β-DAEAs (**2m-az**) from the corresponding aryl acetonitriles: substrate (0.15 mmol), in a mixed solution of 0.5 M K₂CO₃ in D₂O/dioxane (3:1 v/v, 8 mL), LC-Fe cathode (working area: 1.0 cm²), at −1.2 V, RT, 8 h (−1.5 V and −1.4 V are required for **2p** and **2ae**, respectively, and −1.3 V is required for **2x**, **2ab**, and **2ad**). **b** Parallel synthesis of multiple α,β-DAEAs. Reaction conditions: the same amount (0.1 mmol) of **1a**, **1aaa**, and **1al**, a mixed solution of 0.5 M K₂CO₃ in D₂O/dioxane (3:1 v/v, 8 mL), at −1.25 V, RT, 16 h.

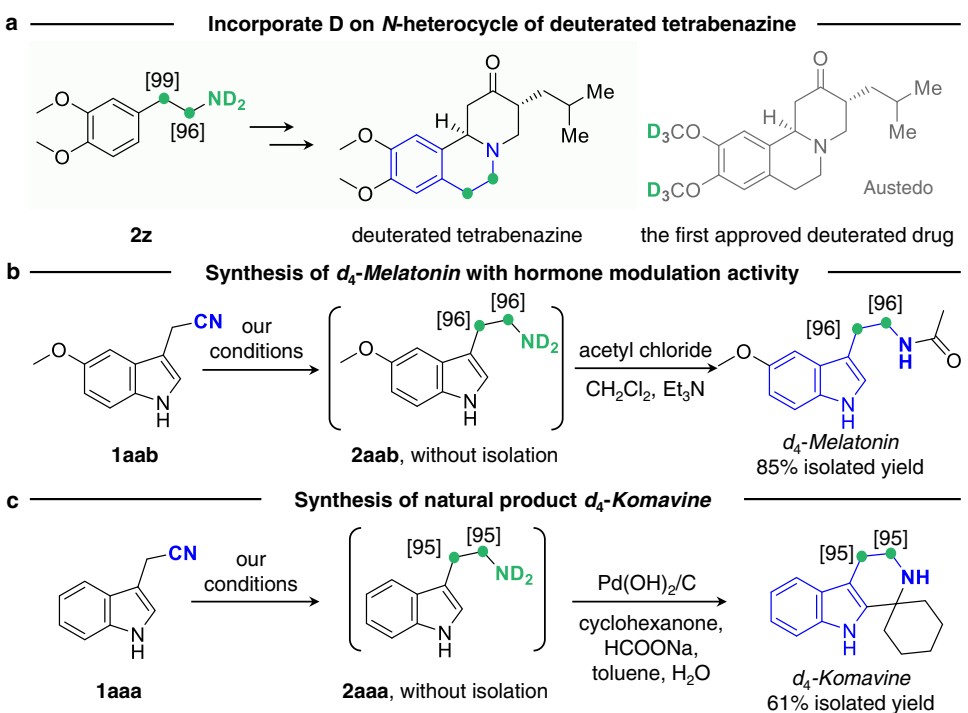

**Fig. 6 | Potential utility for synthesizing deuterated drugs from obtained α,β-DAEAs. a** An alternative route to complement the first US FDA-approved deuterated drug Austedo from α,β-deuterio aryl ethylamine **2z. b** and **c** Examples of using α,β-DAEAs for the synthesis of deuterated drug and natural product.

reactor, highlighting the good promise. Our work not only offers an efficient and promising alternative for the synthesis of α,β-DAEAs under ambient conditions but also offers a paradigm for designing and synthesizing low-coordinated materials to improve the reaction activity and product selectivity for other organic electrocatalytic transformations.

## Methods

### Synthesis of CP-supported α-Fe₂O₃ NRs

The α-Fe₂O₃/CP NRs were synthesized according to a reported method[57]. Typically, 0.17 g FeCl₃·6H₂O and 0.1 g Na₂SO₄ were mixed into 15 mL deionized (DI) water and stirred to obtain a homogenous solution. Then, the mixture was transferred to a Teflon-lined stainless autoclave. A piece of CP (2.0 × 3.0 cm²) was immersed into the solution, and the autoclave was maintained at 120 °C for 6 h in an oven. The obtained FeOOH/CP was washed with DI water several times and dried naturally. Subsequently, the sample was heated to 450 °C for 3 h in a temperature-controlled oven with a heating rate of 3 °C min⁻¹ under an Ar atmosphere, and then naturally cooled to ambient temperature obtaining the α-Fe₂O₃/CP NRs.

### Synthesis of low coordinated LC-Fe/CP NPs

The Ivium-n-Stat electrochemical workstation (Ivium Technologies B.V.) was used for the electroreduction of α-Fe₂O₃/CP NRs to LC-Fe/CP NPs. The in situ electroreduction of α-Fe₂O₃/CP NRs was performed in 1.0 M KOH in a divided three-electrode electrochemical cell consisting of a working electrode, a carbon rod counter electrode, and a Hg/HgO (1.0 M KOH) reference electrode at −1.9 V vs. Hg/HgO. The exposure area of the working electrode was 1.0 cm⁻². The LC-Fe/CP NPs were obtained after 35 mins of electrolysis.

### Characterizations

The morphology of the catalysts was observed by a FEI Apreo S LoVac scanning electron microscope (SEM) with an accelerating voltage of 10.0 kV. The X-ray diffraction (XRD) patterns were analyzed in the range of 10° to 90° at a scan rate of 20° min⁻¹ using a Rigaku Smartlab

9KW diffraction system with a Cu Kα source (λ = 1.54056 Å). The X-ray photoelectron spectra (XPS) measurements were performed on a Thermo Fisher ESCALAB-250Xi spectrometer using a monochromatic Al Kα x-ray beam (1486.60 eV). All the peaks were calibrated by the binding energy of 284.8 eV of the C 1s spectrum. The X-ray absorption spectroscopy (XAS) of the Fe K-edge was undertaken under an ultra-high vacuum at the 1W1B beamline of the Beijing Synchrotron Radiation Facility (BSRF). The XAS spectra were analyzed with the ATHENA software package[48] The NMR spectra were recorded on JEOL JNM-ECZ400S/L1 instrument at 400 MHz (¹H NMR) and 101 MHz (¹³C NMR) with DMSO-$d_6$, CDCl₃, or CD₃CN as the solvents. Chemical shifts were reported in parts per million (ppm) downfield from internal tetra-methylsilane. Multiplicity was indicated as follows: s (singlet), d (doublet), t (triplet), m (multiplet), br (broad). Coupling constants were reported in hertz (Hz). The quantitative analysis of the liquid products was conducted by the gas chromatograph (GC, Agilent 7890 A) with thermal conductivity (TCD), flame ionization detector (FID), and HP-5MS capillary column (0.25 mm in diameter, 30 m in length). Identification of the reactants and products was performed using gas chromatography-mass spectrometry (Agilent, 8860GC-5977MS) with HP-5MS capillary column (0.25 mm in diameter, 30 m in length). The injection temperature was set at 300 °C. Nitrogen was used as the carrier gas at 1.5 mL min⁻¹. Accurate mass measurements of products were obtained via high-resolution mass spectrometry (HR-MS, ESI, positive mode) on an UltrafleXtreme MALDI-TOF mass spectrometer (Bruker Daltonics) or an Agilent 6550 QTOF. Hydrogen and carbon radicals were investigated with electron spin resonance (ESR) spectroscopy (JES-FA200, JEOL, Japan). Key parameters are as follows: field sweep = 3205–3355 G, field modulation frequency = 100 kHz, sweep time = 1 min, microfrequency = 922 GHz, and power = 4.0 mW.

### General procedures for electrochemical measurements

The electrocatalytic hydrogenation of nitriles was carried out in a divided three-electrode electrochemical cell separated by a Nafion 117 proton exchange membrane containing a working electrode (exposure area of 1.0 cm²), a counter electrode (carbon rod), and a Hg/HgO

(1.0 M KOH) reference electrode. 1.0 M KOH/$H_2O$ or 0.5 M $K_2CO_3$/$D_2O$ was employed as electrolyte. After the LC-Fe was in situ formed, 0.1 mmol of substrates dissolved in dioxane were rapidly added to the cathode. Then, chronoamperometry was carried out at a given constant potential under magnetic stirring (600 rpm). After that, the products at the cathode were extracted with dichloromethane (DCM). The DCM phase was removed, and the residuals were analyzed by GC to provide the GC conversion yields. The organic phase was treated with a 3.0 M solution of HCl in cyclopentylmethyl ether, and the precipitated solid products were filtered for calculating the isolated products and with NMR tests. All the potentials in this work were referred to Hg/HgO without *iR* correction unless otherwise stated. All experiments were carried out at room temperature. Galvanostatic reduction of nitriles was also performed in the same divided electrochemical cell.

## In situ Raman spectroscopy measurements

The in situ electrochemical Raman spectroscopy was performed using time-dependent in situ methods on a Renishaw inVia reflex Raman microscope using an excitation of 532 nm (for LC-Fe catalysts) or 633 nm (for organic molecules) laser under controlled potentials by an electrochemical workstation. The electrolytic cell was homemade by Teflon with a piece of round quartz glass as the cover. The working electrode was set to keep the plane of the sample perpendicular to the incident laser. Pt wire was used as the counter-electrode, and Hg/HgO with an internal reference electrolyte of 1.0 M KOH was used as the reference electrode.

## Electrochemical in situ XAS measurements

The in situ electrochemical XAS at the Fe K-edge was recorded at a 1W1B beamline of the BSRF. The electrolytic cell was homemade by Teflon with a Pt plate as the counter-electrode and a Hg/HgO electrode as the reference electrode. The $\alpha$-$Fe_2O_3$/CP NRs were pre-reduced at −1.9 V for 35 minutes, and then the potential was changed to −1.2 V for the XAS measurements for reducing the influence of $H_2$ bubbles for the XAS tests. The XAS spectra were analyzed with the Athena software package.

## Data availability

The data that support the plots within this paper are available from the corresponding author upon reasonable request. The source data underlying Figs. 2–4 are provided as a Source Data file. Source data are provided with this paper.

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

## Acknowledgements

The authors are grateful to the National Natural Science Foundation of China (Nos. 21871206 (B.Z.) and 22001192 (C.L.)). We also acknowledge Dr. Yanmei Shi for her kind help with electrochemical in situ Raman and XRD measurements. We appreciate Dr. Shibo Xi at the XAFCA beamline of the Singapore Synchrotron Light Source and Dr. Lirong Zheng at the 1W1B beamline of the Beijing Synchrotron Radiation Facility for the XAS discussion and support.

## Author contributions

B.Z. and C.L. conceived the idea and directed the research. R.L., C.L., and B.Z. designed the experiments. R.L. and Y.W. synthesized the materials and carried out the electrochemical experiments. R.L., Y.W., and C.L. analyzed the NMR data. C.W. contributed to the density functional theory calculations. M.H. helped to isolate the products and do the NMR tests. C.L. wrote the paper. B.Z. revised the paper with comments from all authors.

## Competing interests

The authors declare no competing interests.
