## [Peer Review File · Nature Communications]

One-pot H/D exchange and low-coordinated iron electrocatalyzed deuteration of nitriles in D₂O to α,β -deuterio aryl ethylaminesEditorial Note: This manuscript has been previously reviewed at another journal that is not operating a transparent peer review scheme. This document only contains reviewer comments and rebuttal letters for versions considered at *Nature Communications* .

REVIEWERS' COMMENTS

Reviewer #1 (Remarks to the Author):

This revised version of the manuscript by Zhang et al. reports an important contribution to the preparation of deuterated α,β -deuterio aryl ethylamines using an iron electrocatalyzed deuteration of nitriles in D₂O.

Previous comments have been well addressed. In addition, the authors have added new data and corrected several overstatements and inconsistencies in the revised manuscript.

I'm very much in favour of this work, but must bring forward a few comments, criticisms, and would like to recommend some changes and corrections that mainly relate to the deuterium content.

From the examples that are given in the SI pages S118...S122...S126...S128., a closer look reveals that the deuterium content is not always calculated correctly. Just as an example, on page S122, some H integrals show 0.67, 0.91 and 0.93 rather than 1. This would mean that at least 33%, 9% and 7% of deuterium was incorporated into the aromatic ring. The same uncertainties can be seen in the Supplementary Information.

Overall, I think this work is worth to be published in Nature Communications.